# Design, Synthesis, Antibacterial, and Antitumor Activity of Linear Polyisocyanide Quaternary Ammonium Salts with Different Structures and Chain Lengths

**DOI:** 10.3390/molecules26185686

**Published:** 2021-09-19

**Authors:** Hongguang Zhang, Lijia Liu, Peng Hou, Jun Liu, Shuang Fu

**Affiliations:** 1College of Pharmacy, Qiqihar Medical University, Qiqihar 161006, China; zhanghg@qmu.edu.cn (H.Z.); houp@qmu.edu.cn (P.H.); L_j2016@qmu.edu.cn (J.L.); 2Key Laboratory of Superlight Materials & Surface Technology, Ministry of Education, Institute of Advanced Marine Materials, College of Materials Science and Chemical Engineering, Harbin Engineering University, Harbin 150001, China; liulijia@hrbeu.edu.cn

**Keywords:** controlled polymerization, polyisocyanide, quaternary ammonium salt, antibacterial, antitumor

## Abstract

The development of organic polymer materials for disinfection and sterilization is thought of as one of the most promising avenues to solve the growth and spread of harmful microorganisms. Here, a series of linear polyisocyanide quaternary ammonium salts (L-PQASs) with different structures and chain lengths were designed and synthesized by polymerization of phenyl isocyanide monomer containing a 4-chloro-1-butyl side chain followed by quaternary amination salinization. The resultant compounds were characterized by ^1^H NMR and FT-IR. The antibacterial activity of L-PQASs with different structures and chain lengths against *Escherichia coli* (*E. coli**)* and *Staphylococcus aureus* (*S. aureus**)* was evaluated by determining the minimum inhibitory concentrations (MICs). The L-POcQAS-M_50_ has the strongest antimicrobial activity with MICs of 27 μg/mL against *E. coli* and 32 μg/mL against *S. aureus.* When the L-PQASs had the same polymerization degree, the order of the antibacterial activity of the L-PQASs was L-POcQAS-M_n_ > L-PBuQAS-M_n_ > L-PBnQAS-M_n_ > L-PDBQAS-M_n_ (linear, polyisocyanide quaternary ammonium salt, monomer, *n* = 50,100). However, when L-PQASs had the same side chain, the antibacterial activity reduced with the increase of the molecular weight of the main chain. These results demonstrated that the antibacterial activity of L-PQASs was dependent on the structure of the main chain and the length of the side chain. In addition, we also found that the L-POcQAS-M_50_ had a significant killing effect on MK-28 gastric cancer cells.

## 1. Introduction

Nowadays, microbial infection is still a heavy threat to public health worldwide. Especially the common bacteria in the environment seriously endanger human health all the time [1,2]. Although using antibiotics is one of the most attractive and effective ways to inhibit the growth of bacteria or directly kill pathogens [3], the harm caused by the residual antibiotics to humans is a new problem that needs to be solved urgently [4]. Hence, the development of harmless antibacterial agents has become a hot focus of research [5].

Quaternary ammonium salts (QASs) are a class of extensively studied materials owing to its long-term antibacterial property, low toxicity, low permeability, and excellent safety [6]. In addition, QASs also show the advantages of designable molecular structure, high functionality, no volatility, prolonged lifetime, and chemical stability [7,8,9]. Hence, they have been extensively applied in the fields of antibacterial and antitumor materials [10,11,12]. For example, some review articles have summarized the relationship between the antibacterial properties of quaternary ammonium salts and its structure [13,14,15]. Furthermore, Linda et al. developed a series of quaternary ammonium compounds (QACs) derived from quinuclidin-3-ol and an alkyl chain of variable length, and found that it is necessary for new compounds to get better antimicrobial potentials by the addition of alkyl chains containing at least ten carbon atoms [16]. Stoikova et al. designed new bactericidal and antibacterial agents based on QACs on thiacalixarene macrocyclic platform. These agents exhibited high antibacterial effect [17]. Yang et al. prepared a series of novel diosgenyl quaternary ammonium derivatives, and found that some derivatives had similar antitumor activity as that of Adriamycin and Tae226 [18]. However, exploring the existing literature, we found that there are a few studies reporting on the use of polyisocyanide as the main chain to design QASs materials.

Recent reports indicate that the antimicrobial and antitumor activity of QASs can be influenced not only by the structure but also by the hydrophilicity and hydrophobicity of QASs [19,20,21]. Hence, it is a feasible strategy to develop new QAS materials with excellent antimicrobial and antitumor activity by changing their structures and chain lengths. In this paper, we synthesized eight kinds of L-PQASs with different main chain structures and side chain lengths using polyisocyanide as the main chain, named as L-PQAS-M_n_ (linear, polyisocyanide quaternary ammonium salt, monomer, *n* = 50,100). The antimicrobial effect of the L-PQAS-M_n_ against *E. coli* and *S. aureus* were examined by determining the minimum inhibitory concentrations (MICs). The correlation between the antimicrobial activities of L-PQASs and their structures and chain lengths was explored. Finally, we also investigated the killing effect of the L-POcQAS-M_50_ on MK-28 gastric cancer cells.

## 2. Results

### 2.1. Structural Characteristics

Generally, the catalytic materials used in the polymerization of isonitrile included nickel catalyst [22], palladium-platinum binary catalyst [23], and rhodium catalyst [24]. However, the molecular weight distribution of the polymer obtained by the polymerization of isonitrile is wide when we used nickel as catalyst, and the length of the polymer chain is not easy to control. Although the Pd(II)-Pt(II) binary catalyst can be used to obtain polymers with narrow molecular weight distribution, it is very difficult to prepare the catalyst because it is unstable and sensitive to air. Furthermore, the price of rhodium metal is very expensive and its synthesis is more difficult. Based on the above considerations, we chose a palladium catalyst with simple synthetic method, stable properties, and high initiation efficiency to initiate the polymerization of isonitrile. To prove the successful synthesis of L-P-M_n_ (*n* = 50,100), SEC analysis of the prepared polymers was carried out (Figure 1). Catalyst-initiated polymerizations of monomer in different initial feed ratios (n) were performed under the same conditions ([Monomer]_0_ = 0.2 M, [Monomer]_0_/[Catalyst]_0_ = n, THF, 55 °C). It can be seen that the molecular weight of L-P-M_n_ gradually increases with the increase of the initial monomer and catalyst concentration (Figure 1). Furthermore, we can see that the number-averaged molecular masses (M_n_) and the polydispersity indices (M_w_/M_n_) of L-P-M_50_ and L-P-M_100_ are as follows: L-P-M_50_ (M_n_ = 10,125, M_w_/M_n_ = 1.12) and L-P-M_100_ (M_n_ = 21,310, M_w_/M_n_ = 1.09) (Table 1). These results indicated that we successfully prepared L-P-M_n_ (*n* = 50,100) with different molecular weights and narrow molecular weight distributions.

^1^H NMR and FT-IR were used to characterize the structures of L-P-M_50_ and L-PQASs. Appendix A showed the ^1^H NMR spectrum of the L-P-M_50_, we can observe a series of characteristic resonances assignable to the L-P-M_50_ structure, including those of the phenyl protons (at 7.33 ppm and at 5.71 ppm), and chlorobutyl protons (at 3.89–4.06 ppm, at 3.51 ppm and at 1.78–1.90 ppm). Appendix A showed the ^1^H NMR spectrum of the L-PBnQAS-M_50_. Compared to the ^1^H NMR spectrum of the L-P-M_50_, two new peaks are observed at 4.76 ppm and 3.11 ppm, which were assigned to the hydrogen protons of the methylene group on the benzyl group (Ar-CH_2_) and the hydrogen protons of the two methyl groups on the quaternary ammonium salt structure (N^+^(CH_3_)_2_), respectively. Similarly, the characteristic peaks of hydrogen protons on the corresponding quaternary ammonium salt structure appeared in the ^1^H NMR of the L-PBuQAS-M_50_, L-POcQAS-M_50_, and L-PDBQAS-M_50_ (Appendix A), which indicated that the L-PQASs had been successfully obtained.

The successful synthesis of L-PQASs was further evidenced by the comparison studies of FT-IR spectra (Figure 2). Compared to the FT-IR spectrum of L-P-M_50_ (control) in Figure 2a, the four L-PQASs showed new peak at 3068 cm^−1^, 2880 cm^−1^, 2886 cm^−1^, and 2878 cm^−1^, respectively (Figure 2b–e), which was assigned to the absorption peak of the C-H stretching vibration in the introduced quaternary ammonium group. Furthermore, in the FT-IR spectra of the four L-PQASs, it can be seen that the absorption peak of C-Cl stretching vibration at 645 cm^−1^ disappeared, indicating the L-P-M_50_ successfully achieved quaternization. These experiment data further confirmed that the L-PQASs had been successfully fabricated.

### 2.2. Solubility and Antimicrobial Activity

Many literatures have reported that the water solubility of the QAS is one of the important factors affecting its antimicrobial activity. In this article, we dissolve 10 mg of polyisocyanide quaternary ammonium salts in 1.00 mL of deionized water to evaluate their solubility. As shown in Table 2, we can see that L-PBuQAS-M_50_, L-POcQAS-M_50_, and L-POcQAS-M_100_ have high water solubility, but L-PDBQAS-M_50_, L-PBnQAS-M_100_, and L-PDBQAS-M_100_ have low water solubility. Generally speaking, the better water solubility QASs have, the higher antibacterial activity they have.

Furthermore, the antimicrobial activities of eight L-PQASs were evaluated by in vitro minimum inhibitory concentrations (MICs) (Table 2). As clearly seen in Table 2, the synthesized L-PQASs have MICs in the range of 27 to 500 µg/mL against *E. coli* and *S. aureus*. It is worth noting that L-POcQAS-M_50_ exhibited the highest antimicrobial activity, and the MIC of L-POcQAS-M_50_ was 27 µg/mL on *E. coli*, and 32 µg/mL on *S. aureus*, respectively. In addition, when the main chain was the same, we found that the antimicrobial activities of the as-prepared L-PQASs-M_50_ were observed in the following order: L-POcQAS-M_50_ > L-PBuQAS-M_50_ > L-PBnQAS-M_50_ > L-PDBQAS-M_50_.

### 2.3. The Relationship between Structure and Antibacterial Activity

The structure of QASs and the length of alkyl chains are crucial to its antibacterial activity. As we know, the surface of bacterial cell is negatively charged. When the bacteria contacts the L-PQASs, the longer hydrophobic alkyl chain in the L-PQASs structure will pierce the cell wall of the bacteria. Subsequently, the QAS group with positive charge activates the contact killing mechanism to kill bacteria due to electrostatic interaction. The shorter the alkyl chain of QASs is, the weaker the ability to pierce the cell wall. Similar result has been reported in previous literatures [8,25]. Therefore, L-PBnQAS-M_50_ with shorter alkyl chains exhibited poor antibacterial activity. However, we found that the antibacterial ability of L-PDBQAS-M_50_ is much worse than that of the L-POcQAS-M_50_, which could be attributed to its poor water solubility caused by the excessively long alkyl chain. We can conclude that the optimum side chain of L-PQASs is octyl. However, when the side chain is the same, it can be seen that the antibacterial ability of L-PQAS-M_50_ is higher than that of the L-PQAS-M_100_. The MIC of L-POcQAS-M_50_ was 27 µg/mL on *E. coli*, but the MIC of L-POcQAS-M_100_ was 33 µg/mL on *E. coli*. This conclusion indicates that the structure of the L-PQASs is important for its antibacterial performance. In particular, the antibacterial abilities of L-PBnQAS-M_100_ and L-PDBQAS-M_100_ were very weak and almost insignificant. It may be due to the fact that the water solubility of L-PBnQAS-M_100_ and L-PDBQAS-M_100_ are worse than that of L-PBnQAS-M_50_ and L-PDBQAS-M_50_. These results indicate that the suitable alkyl chain length and the excellent water solubility of L-PQASs are the key factors for its antibacterial properties.

### 2.4. Antitumor Bioactivity

Based on the excellent antimicrobial activity of L-POcQAS-M_50_, we further studied its antitumor activity. The antitumor activity of L-POcQAS-M_50_ was evaluated by MTT assay. As shown in Figure 3, L-POcQAS-M_50_ showed promising antitumor activity toward MK-28 gastric cancer cells with IC_50_ of 103.67 µg/mL, and survival of MK-28 gastric cancer cells decreased significantly with the increase of L-POcQAS-M_50_ concentration. When the concentration of L-POcQAS-M_50_ was 200 µg/mL, almost all normal MK-28 gastric cancer cells were found to lose viability (Figure 3c). We deduced that the high water solubility and positive charge are the reasons why the L-POcQAS-M_50_ has excellent antitumor activity.

## 3. Materials and Methods

### 3.1. Materials

Dichlorobis(triethylphosphine)-palladium(II) (Pd(PEt_3_)_2_Cl_2_) (CAS: 28425-04-9), cuprous chloride (CuCl) (CAS: 7758-89-6), dichloromethane (DCM) (CAS: 75-09-2), triethylamine (NEt_3_) (CAS: 121-44-8), phenylacetylene (C_8_H_6_) (CAS: 536-74-3), N-hexane (C_6_H_14_) (CAS: 110-54-3), formic acid (CH_2_O_2_) (CAS: 64-18-6), acetic anhydride (C_4_H_6_O_3_) (CAS: 108-24-7), *p*-aminobenzoic acid (C_7_H_7_NO_2_) (CAS: 150-13-0), 1-(3-dimethylaminopropyl)-3-ethylcarbodiimide hydrochloride (EDCI) (CAS: 25952-53-8), 4-dimethylaminopyridine (DMAP) (CAS: 1122-58-3), 4-chloro-1-butanol (C_4_H_9_ClO) (CAS: 928-51-8), anhydrous magnesium sulfate (MgSO_4_) (CAS: 139939-75-6), iodine (I_2_) (CAS: 7553-56-2), triphenylphosphine (PPh_3_) (CAS: 603-35-0), sodium bicarbonate (NaHCO_3_) (CAS: 144-55-8), *N*,*N*-dimethylformamide (DMF) (CAS: 68-12-2), *N*,*N*-dimethylbenzylamine (BDMA) (CAS: 103-83-3), ethyl acetate (C_4_H_8_O_2_) (CAS: 141-78-6), *N*,*N*-dimethylbutylamine (C_6_H_15_N) (CAS: 927-62-8), *N*,*N*-dimethyloctylamine (C_10_H_23_N) (CAS: 7378-99-6), and *N*,*N*-dimethyldodecylamine (C_14_H_31_N) (CAS: 112-18-5) were purchased from Sigma-Aldrich. *E. coli* (ATCC25922) and *S. aureus* (ATCC25923) were obtained from the first affiliated hospital of Harbin Medical University (Harbin, China). All solvents were used as supplied without additional purification.

### 3.2. Synthesis of Palladium Catalyst

The synthesis of the palladium catalyst is carried out according to the experimental procedure of Wu et al. [26]. In a typical synthesis (Figure 1), 0.41 g (1.02 mmol) of Pd(PEt_3_)_2_Cl_2_ and 6.0 mg (0.06 mmol) of CuCl were added into a 50 mL two-necked flask and the flask was purged three times with dry nitrogen and sealed under vacuum. Then, 10.0 mL of CH_2_Cl_2_, 5.0 mL of NEt_3_, and 130 µL (1.20 mmol) of C_8_H_6_ were added to the mixture under dark and N_2_ condition. After continuous stirring for 12 h at room temperature, the substrate was detected by TLC analysis. When the substrate was almost completely consumed, the solvent was removed and column chromatography was used to purify the solid. Subsequently, the product was dissolved in a small amount of DCM, followed by the addition of appropriate amount of C_6_H_14_ to recrystallize in a refrigerator. Finally, 0.32 g of light yellow crystals was obtained (74.8% yield).

Spectroscopic data of palladium catalyst: ^1^H NMR (DMSO-d_6_, 25 °C, 500 MHz): δ 7.15–7.26 (m, 5H, Ph-*H*), 1.87–1.93 (m, 12H, P-C*H*_2_CH_3_), 1.10–1.16 (m, 18H, P-CH_2_C*H*_3_); LC-MS (ESI) *m*/*z*: [M − H]^−^ Calcd for C_20_H_37_ClP_2_Pd_3_ 481.33; Found 482.11.

### 3.3. Synthesis of Isonitrile Monomer

The overall synthetic approach is reported in Figure 2.

#### 3.3.1. Synthesis of Intermediate **a**

First, 3.8 mL (0.10 mol) of HCOOH was added into a 100 mL flask, and 9.8 mL (0.10 mol) of (CH_3_CO)_2_O was added dropwise with stirring. Then the mixture was heated to 55 °C for 3 h and cooled to room temperature to obtain methyl acetic anhydride. Subsequently, in a 100 mL two-necked flask, 4.11 g (30.00 mmol) of C_7_H_7_NO_2_ was dissolved in 60 mL of DCM until a uniform suspension was obtained. The obtained methyl acetic anhydride was then added dropwise to the suspension in an ice–water bath. When the addition was complete, the ice bath was removed, and vigorously stirred for 12 h. Finally, the solvent was spin-dried under reduced pressure and 4.70 g of white solid **a** was obtained (95.1% yield, m.p. 200.6 °C). The resulting products did not require purification.

Spectroscopic data of intermediate **a**: ^1^H NMR (DMSO-d_6_, 25 °C, 500 MHz): δ 12.68 (s, 1H, COO*H*), 10.40–10.47 (m, 1H, N*H*), 8.52–8.97 (m, 1H, C*H*O), 7.26–8.32 (m, 5H, Ar-*H*); LC-MS (ESI) *m*/*z*: [M − H]^−^ Calcd for C_8_H_7_NO_3_ 165.15; found 165.04.

#### 3.3.2. Synthesis of Intermediate **b**

First, 3.32 g (20.00 mmol) of intermediate **a**, 4.60 g (24.00 mmol) of EDCI, and 1.95 g (16.00 mmol) of DMAP were added into a 100 mL two-necked flask. Subsequently, 50.0 mL of DCM and 2.9 mL (30.00 mmol) of C_4_H_9_ClO were added to the flask. The mixture was stirred until the starting material was completely consumed as judged by TLC. The organic phase was washed with deionized water (50.0 mL × 2) and saturated aqueous NaHCO_3_ (50.0 mL × 2), then dried with magnesium sulfate and the solvent was removed using a rotary evaporator. Finally, the mixture was further purified by column chromatography (silica gel, petroleum ether: ethyl acetate = 4:1 *v*/*v*) to give the intermediate **b** (4.14 g, yield 81.2%, m.p. 157.4 °C) as a white solid.

Spectroscopic data of intermediate **b**: ^1^H NMR (CDCl_3_, 25 °C, 500 MHz): δ 8.44–8.87 (d, *J* = 1.5 Hz, 1H, C*H*O), 8.05 (s, 1H, N*H*), 7.64–7.66 (d, *J* = 3.5 Hz, 2H, Ph-*H*), 7.15–7.16 (d, *J* = 7.1 Hz, 2H, Ar-*H*), 4.34–4.36 (t, 2H, OC*H*_2_), 3.62–3.64 (t, 2H, CH_2_C*H*_2_Cl), 1.94–1.96 (m, 4H, CH_2_C*H*_2_C*H*_2_CH_2_Cl); LC-MS (ESI) *m*/*z*: [M − H]^−^ Calcd for C_12_H_14_ClNO_3_ 255.70; Found 255.07. IR (KBr, cm^−1^): 3307 (*ν*_N-H_), 1527 (amide I), 1287 (amide II), 1708 (*ν*_C=O_ ester), 1606 (*ν*_C=C_ benzene ring), 1302 (*ν*_C-H_ methylene), 1179 (*ν*_C-O_ ester).

#### 3.3.3. Synthesis of Isonitrile Monomer

First, 2.55 g (10.00 mmol) of intermediate **b**, 3.81 g (15.00 mmol) of I_2_, 3.93 g (15.00 mmol) of PPh_3_, and 30.0 mL of DCM were added into a 100 mL two-necked flask. Subsequently, 4.2 mL (30.05 mmol) of NEt_3_ was added dropwise with stirring into an ice–water bath. After continuous stirring for 1 h at room temperature, the substrate **b** was judged to be completely consumed by TLC. Ice-cold water (50 mL) was added immediately to quench the reaction. The organic phase was washed with deionized water (50.0 mL × 2) and saturated aqueous NaHCO_3_ (50.0 mL × 2), then dried with magnesium sulfate and the solvent was removed using a rotary evaporator. The mixture was further purified by column chromatography (silica gel, petroleum ether : ethyl acetate = 4:1 *v*/*v*) to give the crude product. Finally, the crude product was recrystallized from DCM/hexanes as a white solid (1.51 g, 63.7% yield).

Spectroscopic data of isonitrile monomer: ^1^H NMR (CDCl_3_, 25 °C, 500 MHz): δ 8.06–8.08 (d, *J* = 10.5 Hz, 2H, Ph-*H*), 7.44–7.46 (d, *J* = 10.5 Hz, 2H, Ar-*H*), 4.34–4.36 (t, 2H, OC*H*_2_), 3.59–3.62 (t, 2H, CH_2_C*H*_2_Cl), 1.94–1.96 (m, 4H, CH_2_C*H*_2_C*H*_2_CH_2_Cl); LC-MS (ESI) *m*/*z*: [M − H]^−^ Calcd for C_12_H_12_ClNO_2_ 237.68; Found 237.06. IR (KBr, cm^−1^): 3207 (*ν*_N-H_), 1527 (amide I), 1287 (amide II), 2125 (*ν*_C≡N_), 1723 (*ν*_C=O_ ester), 1607 (*ν*_C=C_ benzene ring), 1275 (*ν*_C-H_ methylene), 1119 (*ν*_C-O_ ester).

### 3.4. Synthesis of L-PQASs

The synthesis route of targeted products is shown in Figure 3. In detail, a certain amount of isonitrile monomer and palladium catalyst was respectively dissolved in dry and oxygen-free THF to form two solutions containing a certain concentration. Then, appropriate amount of isonitrile monomer solution and palladium catalyst solution were injected into the polymerization reaction tube with a microsyringe. Subsequently, an appropriate amount of dry oxygen-free THF was added to dilute the solution. The initial concentration of isonitrile monomer was 0.2 M, and the initial isonitrile monomer to palladium catalyst molar ratio ([Monomer]_0_/[Catalyst]_0_) was n (*n* = 50,100). The mixture was stirred at 55 °C for 10 h and then cooled to room temperature. Finally, the mixture was precipitated in methanol and washed with methanol (50.0 mL × 3). The obtained yellow solid was dried under vacuum at 50 °C, which was defined as L-P-M_n_ (linear, polyisocyanide, monomer, *n* = 50,100). The yield of L-P-M_50_ and L-P-M_100_ was 88.2% and 92.1%, respectively. Gel permeation chromatography (GPC) was used to determine the molecular weight (*M*_n_) and polydispersity index (PDI) of the polymers.

Then, the L-P-M_n_ was reacted with one of the tertiary amines (*N*,*N*-dimethylbenzylamine, *N*,*N*-dimethylbutylamine, *N, N*-dimethyloctylamine, and *N*,*N*-dimethyldodecylamine) to synthesize the corresponding quaternary ammonium salts. In a typical synthesis process, 50.0 mg of L-P-M_n_ and 10.0 mL of DMF were stirred at room temperature for 5 h. Then, 2.0 mL of *N*,*N*-dimethylbenzylamine was added and continually stirred for 96 h at 80 °C. Subsequently, the mixture was cooled to room temperature. Finally, the mixture was precipitated in ethyl acetate and washed with ethyl acetate. The obtained solid was dried under vacuum, which was defined as L-PBnQAS-M_n_. In the same way, L-PBuQAS-M_n_, L-POcQAS-M_n_, and L-PDBQAS-M_n_ were synthesized by a process similar to that described above.

Spectroscopic data of L-P-M_50_: ^1^H NMR (CDCl_3_, 25 °C, 500 MHz): δ 7.33 (s, 2H, Ph-*H*), 5.71 (s, 2H, Ar-*H*), 3.89–4.06 (d, *J* = 16.5 Hz, 2H, OC*H*_2_), 3.51 (s, 2H, CH_2_C*H*_2_Cl), 1.78–1.90 (m, 4H, CH_2_C*H*_2_C*H*_2_CH_2_Cl). IR (KBr, cm^−1^): 2963 and 1276 (*ν*_C-H_ methylene), 1697 (*ν*_C=O_ ester), 1603 (*ν*_C=N_), 856 (1,4-disubstituted benzene), 770 (*ν*_C-H_ -(CH_2_)_n_-), 645 (*ν*_C-Cl_).

Spectroscopic data of L-PBnQAS-M_50_: 1H NMR (CDCl_3_, 25 °C, 500 MHz): δ 7.47 (s, 2H, Ph-*H*), 5.74 (s, 2H, Ar-*H*), 4.76 (s, 2H, Ar-C*H*_2_), 3.11 (s, 6H, N^+^(C*H*_3_)_2_), the other characteristic peaks are consistent with those of L-P-M_50_. IR (KBr, cm^−1^): 3068 (*ν*_C-H_ benzylamine ring), 2963 and 1276 (*ν*_C-H_ methylene), 1697 (*ν*_C=O_ ester), 1603 (*ν*_C=N_), 856 (1,4-disubstituted benzene), 770 (*ν*_C-H_ -(CH_2_)_n_-).

Spectroscopic data of L-PBuQAS-M_50_: ^1^H NMR (CDCl_3_, 25 °C, 500 MHz): δ 2.98–3.12 (m, 10H, C*H*_2_-N^+^(C*H*_3_)_2_-C*H*_2_), 1.34 (s, 2H, C*H*_2_CH_3_), 0.88 (s, 3H, N^+^-(CH_2_)_3_-C*H*_3_), the other characteristic peaks are consistent with those of L-P-M_50_. IR (KBr, cm^−1^): 2880 (*ν*_C-H_ methyl in butyl), 2963 and 1276 (*ν*_C-H_ methylene), 1697 (*ν*_C=O_ ester), 1603 (*ν*_C=N_), 856 (1,4-disubstituted benzene), 770 (*ν*_C-H_ -(CH_2_)_n_-).

Spectroscopic data of L-POcQAS-M_50_: ^1^H NMR (CDCl_3_, 25 °C, 500 MHz): δ 3.17 (m, 6H, N^+^(C*H*_3_)_2_), 1.27 (s, 14H, (C*H*_2_)_7_CH_3_), 0.86 (s, 3H, (CH_2_)_7_C*H*_3_), the other characteristic peaks are consistent with those of L-P-M_50_. IR (KBr, cm^−1^): 2886 (*ν*_C-H_ methyl in n-octyl), 2963 and 1276 (*ν*_C-H_ methylene), 1697 (*ν*_C=O_ ester), 1603 (*ν*_C=N_), 856 (1,4-disubstituted benzene), 770 (*ν*_C-H_ -(CH_2_)_n_-).

Spectroscopic data of L-PDBQAS-M_50_: ^1^H NMR (CDCl_3_, 25 °C, 500 MHz): δ 3.15 (m, 6H, N^+^(C*H*_3_)_2_), 1.26 (s, 22H, (C*H*_2_)_11_CH_3_), 0.86 (s, 3H, (CH_2_)_11_C*H*_3_), the other characteristic peaks are consistent with those of L-P-M_50_. IR (KBr, cm^−1^): 2878 (*ν*_C-H_ methyl in dodecyl), 2963 and 1276 (*ν*_C-H_ methylene), 1697 (*ν*_C=O_ ester), 1603 (*ν*_C=N_), 856 (1,4-disubstituted benzene), 770 (*ν*_C-H_ -(CH_2_)_n_-).

### 3.5. Antibacterial Tests

Antimicrobial activities of L-PQASs against *E. coli* and *S. aureus* were evaluated by determining the minimum inhibitory concentrations (MICs). Bacterial suspension after overnight culture was diluted to 10^6^~10^7^ CFU/mL. Eight kinds of L-PQASs were separately dissolved in sterile distilled water to make a concentration of 20 mg/mL. Total of 100 µL of bacterial culture including 0.02% TTC was added to each well of a 96-well plate. Subsequently, 100 µL of L-PQASs were two-fold serially diluted with broth in a 96-well microplate for each bacteria, resulting in final concentrations from 5000 to 2.5 µg/mL. A clear nutrient broth was used as a negative control and a nutrient broth with the bacterial culture was used as the positive control. Finally, the plates were covered and incubated at 37 °C for 24 h.

### 3.6. Antitumor Bioactivity Tests

The MTT assay was performed to evaluate the antitumor activity of the L-POcQAS-M_50_. MK-28 gastric cancer cells were obtained from the research institute of medicine and pharmacy, Qiqihar Medical University (Qiqihar, China). First, MK-28 gastric cancer cells were seeded onto a 96-well plate and cultured for 24 h at 37 °C. Next, different concentrations of the L-POcQAS-M_50_ (200, 100, 50, 25, 12.5, 6.25, and 0 µg/mL) were added, and the cells were cultured for another 24 h. Subsequently, 20 μL of the MTT solution (5 mg/mL in PBS) was added to each well and incubated with the cells for 4 h. After removing the medium, 150 μL of DMSO was added per well. The percentage of cell viability was measured on a microplate reader at 490 nm.

### 3.7. Characterizations

The chemical structures of all samples were verified by Fourier transform infrared (FTIR) spectra (Spectrum 100, PerkinElmer, Waltham, Massachusetts, USA). ^1^H NMR spectra were collected using a Bruker Avance III-500 (500 MHz) spectrometer at ambient temperature using CDCl_3_, or DMSO-d6 as the solvent. The cells were observed under an optical microscope (Chongqing Optical Instrument Factory, Chongqing, China) and recorded.

## 4. Conclusions

In summary, a series of L-PQASs with excellent antibacterial activity was successfully developed using polyisocyanide as the main chain, followed by quaternary amination salinization. According to the characterization results of ^1^H NMR and FT-IR, it can be determined that the resultant compounds contained quaternary ammonium groups. The L-POcQAS-M_50_ containing a long alkyl chain and QAS group showed the satisfactory antimicrobial activity, and the MIC of L-POcQAS-M_50_ to *E. coli* and *S. aureus* was 27 µg/mL and 32 µg/mL, respectively. L-POcQAS-M_50_ with long alkyl chain have good water solubility, which meant it easily contacts the bacteria and disrupts its cell wall. On the other hand, after the introduction of the QAS group with positive charge, the cell membrane of the bacteria will be disrupted, resulting in the death of the bacteria. Furthermore, L-POcQAS-M_50_ has a significant inhibitory effect on the MK-28 gastric cancer cells (IC_50_ = 103.67 µg/mL). With the excellent antimicrobial and antitumor property, there is no doubt that the L-PQASs have a great potential for practical application in medical and health fields.

## Data Availability

Not applicable.

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
