# Peer review of "Design, Synthesis, Antibacterial, and Antitumor Activity of Linear Polyisocyanide Quaternary Ammonium Salts with Different Structures and Chain Lengths"

_molecules, 2021, doi:10.3390/molecules26185686_

Round 1
Reviewer 1 Report
Article (molecules-1365386) entitled “Design, synthesis, antibacterial and antitumor activity of linear polyisocyanate quaternary ammonium salts with different structures and chain lengths by Zhang group reports the synthesis of linear polyisocyanate quaternary ammonium salts with different chain lengths and antibacterial study against two strains against E. coli and S. aureus. Paper was explained scientifically without solid evidence in characterization is my main concern regarding manuscript. Anyway, I recommend paper for publication in Molecules after defending my reservations as shown below:
Abstract
In general, to avoid abbreviation in stains name, E. coli and S. aureus, spell out abbreviations the first time, followed by the acronym in parentheses
Synthesis & Structural characteristics
As authors mentioned the design and synthesis of various linear polyisocyanate quaternary ammonium salts and characterization using only simple FTIR and Proton NMR. I think it is difficult to characterize polymer materials using two simple spectroscopy techniques. How to prove synthesis of (linear, polyisocyanate quaternary ammonium salt, monomer, n=50, 100) without molecular weight, stereochemistry (orientation of side chain). 1H NMR spectra did not adequately describe L-PQASs without mentioning integration in NMR, (provide full NMR of any L-PQASs with integration as a supplementary information). Author should include 13C NMR and elemental analyses or HRMS if possible, to support carbon skeleton and weight. In the experimental section, nature and melting/decomposing points must be included.
Synthesis of palladium catalyst
Catalyst mentioned in paper was reported first time by authors or using reported method or purchased it. If authors have synthesized a catalyst for the first time, provide 31P NMR, elemental analysis, and melting point. Otherwise, to acknowledge the contribution of others, references to the synthesized catalyst should be cited in the text; in addition, the author should compare the melting point of the reported catalyst in your paper with the documented melting point presented elsewhere.
Some other points
Authors found that the L-POcQAS-M50 had a significant killing effect on MK-28 gastric cancer cells but did not conduct experimental studies in normal cells to verify toxicity.????
In the paper, double-check and add proper references.
Reviewer 2 Report
The authors present in this paper the synthesis and characterization of a series of linear polyisocyanate quaternary ammonium salts (L-PQASs), with different structures and chain lengths, via the polymerization of a phenyl isocyanide monomer containing a 4-chloro-1-butyl side chain followed by salinization by quaternary amination. In addition, they present the results of the evaluation of the antibacterial and antiproliferative activities of L-PQAS against E. coli and S. aureus. The presented work remains original since, in terms of plagiarism assessment, only about 8% of this paper consists of texts more or less similar to the content of 135 sources considered as the most relevant. The manuscript is written in a very clear and interesting way. The experimental part and the characterization of the compounds are described in a detailed manner. I found this article interesting and consider the manuscript suitable for publication after minor revision.
For a good organization of the structure of the article, the authors in section 3 relating to materials and methods, are asked to keep only the general procedure for the synthesis of palladium catalyst, intermediates 2 and 3, and isonitrile monomer. As for the section 2 corresponding to the results, the authors are requested to present (without the detail of the reaction protocol) the methods used for the synthesis of the products with their schemes, to discuss the obtained results and to specify the advantage of the polymerization method thus used. It is also recommended to specify in the text the corresponding CAS for commercial products

Reviewer 3 Report
The manuscript molecules-1365386 " Design, synthesis, antibacterial and antitumor activity of linear polyisocyanate quaternary ammonium salts with different structures and chain lengths" by Zhang et.al. describes the synthesis of linear polymeric quaternary ammonium with different structures and chain lengths, and the study of their antimicrobial and antitumor activity. The synthesis of novel compounds was confirmed by NMR and FT-IR spectroscopy.
Comments / suggestions:
1) I recommend that the authors strengthen the Introduction part on the design of new compounds with antimicrobial activity. New review articles on the design of quaternary ammonium salts, as well as their properties, should be added. For example, Molecules 2020, 25(21), 5145, Int. J. Mol. Sci. 2021, 22(13), 6793, Pathogens 2020, 9(6), 459.
2) In the manuscript, the parts are now very illogical and inconsistent, namely, first the part about the characterization of compounds and their properties, and only then the description of their synthesis. I recommend that authors look at Molecules for several synthetic articles with biological activity.
3) Since almost all compounds were obtained for the first time, new compounds should be characterized by few physical methods (1H, 13C NMR, IR spectroscopy and mass spectrometry). Physical methods of polymer chemistry are also needed to characterize polymer products (L-PQASs). Images of all spectra should be added in supplementary materials.
4) Why do the authors call the obtained polymer derivatives a polyisocyanate? The monomer unit is nitrile, not isocyanate.
5) How did the authors evaluated the solubility of the obtained compounds qualitatively and quantitatively?
6) The study does not assess the impact of test compounds on normal cells.
7) The part of the Discussion is poorly presented in the manuscript. I recommend the authors add a discussion about the structure-property relationship for the obtained compounds.
8) The text of the manuscript there is no information about the reproducibility of the results.
9) The manuscript should be rechecked for typos and errors.
10) English must be checked by a native speaker.
Reviewer 4 Report
The article Design, synthesis, antibacterial and antitumor activity of linear polyisocyanate quaternary ammonium salts with different structures and chain lengths described application of quaternary ammonium salts for medicinal purposes. The style and language of the article is acceptable even though I would suggest some improvement of the language style.
I have some comments and remarks:
I do not understand why is in introduction mention COVID-19 which is off topic. Please remove this sentence and related references from the text.
Composition of polymeric material L-P-Mn was tested or not?
Since the distribution of polymers can play a role in biological activity, it would be good to re-test biological activity of final compounds prepared from different batches of L-P-Mn.
How was performed solubility test? What is criteria for solubility?
According to Fig.1, it is impossible to verify the NMR interpretation as is described in Material and Methods.
For example Fig.1-a there should be two doublets (7.33 and 3.89 -4.06).
1) Why the range 3.89-4.06 is showed? The doublet should be described by one frequency and coupling constant?
2) From Fig.1 none of this signals looks as a doublet.
Rewrite NMR according the standards form NMR description (namely coupling constants are missing).
This article can be accepted but I would suggest serious revision.
Reviewer 5 Report
Fu et al reported on the design, synthesis, antibacterial and antitumor activity of linear polyisocyanate quaternary ammonium salts with different structures and chain lengths. It is an interesting piece of paper and with good results in terms of the biological activity of the prepared polymers. Studies on the preparation of these materials by using polyisocyanide as the main chain are scarce and their antimicrobial and antitumor activity are satisfactory to excellent. The manuscript deserves to be published after some changes:
- Page 2, line 45: quinuclidine-3-ol must be quinuclidin-3-ol
- The quality of Figure 1 should be improved in terms of size and resolution. In the current form, it is hard to analyse the 1H NMR spectra presented.
- The authors should also present the C-13 NMR spectra to confirm the structure of the prepared polymers.
- The names of the antimicrobials - coli and S.aureus – must be italicized along all the manuscript
- The synthetic experimental procedure must adapted accordingly to the format of the journal.
- Page 5, line 165: the aromatic signals must be 5H - 7.15-7.26 (m, 5H, Ph-H),
- Page 5, line 172: the authors referred C7H7NO2, but scheme 2 indicate C7H7NH2: Please correct.
- Schemes 2 and 3 must indicate the experimental conditions of each step depicted. It is missed a great number of reagents and the experimental conditions.
- The authors must explain how they purified the compounds. Ex. “The mixture was further purified by column chromatography to give the crude product”. Which was the eluent?
- Structural characterisation of the prepared compounds: NMR – the multiplicity of the signals must be accompanied by the coupling constant.
- In the name of compounds there is some letters that must be italicized: N, p, of p-aminobenzoic acid, N,N-dimethylformamide (DMF), N,N-dimethylbenzylamine (BDMA), N,N-dimethylbutylamine (C6H15N), N,N-dimethyloctylamine (C10H23N), and N,N-dimethyldodecylamine.
Round 2
Reviewer 3 Report
I would like to thank the authors for improving the manuscript.